# Exact Path Kernels Naturally Decompose Model Predictions

## Abstract

This paper proposes a generalized exact path kernel gEPK which naturally decomposes model predictions into localized input gradients or parameter gradients. Many cutting edge out-of-distribution (OOD) detection methods are in effect projections onto a reduced representation of the gEPK parameter gradient subspace. This decomposition is also shown to map the significant modes of variation that define how model predictions depend on training input gradients at arbitrary test points. These local features are independent of architecture and can be directly compared between models. Furthermore this method also allows measurement of signal manifold dimension and can inform theoretically principled methods for OOD detection on pre-trained models.

## 1 Introduction

Out-of-distribution (OOD) detection for machine learning models is a new, quickly growing field important to both reliability and robustness (Hendrycks & Dietterich, 2019; Biggio et al., 2014; Hendrycks & Gimpel, 2017; Silva et al., 2023; Yang et al., 2021; **?**). Recent results have empirically shown that parameter gradients are highly informative for OOD detection (Behpour et al., 2023; **?**; Huang et al., 2021a). To our knowledge, this paper is the first to present theoretical justifications which explain the surprising effectiveness of parameter gradients for OOD detection.

In this paper, we unite empirical insights in cutting edge OOD with recent theoretical development in the representation of finite neural network models with tangent kernels (Bell et al., 2023; Chen et al., 2021b; Domingos, 2020). Both of these bodies of work share approaches for decomposing model predictions in terms of parameter gradients. However, the Exact Path Kernel (EPK) (Bell et al., 2023) provides not only rigorous theoretical foundation for the use of this method for OOD, but also naturally defines other decompositions which deepen and expand our understanding of model predictions. The application of this theory is directly connected to recent state of the art OOD detection methods.

In addition, this paper provides a connection between tangent kernel methods and dimension estimation. At the core of this technique is the ability to extract individual training point sensitivities on test predictions. This paper demonstrates a generalization (the gEPK) of the EPK from Bell et al. (2023), which can exactly measure the *input gradient* $\nabla_{x_{\text{train}}} f(x_{\text{test}}; \theta_{\text{trained}})$. It is shown that this quantity provides all necessary information for measuring the dimension of the *signal manifold* Srinivas et al. (2023) around a given test point.

In short, this work leverages the gEPK to:

- Generalize and explain the success of recent successful methods in OOD.
- Showcase OOD using natural gEPK based decomposition of model predictions in terms of parameter gradients.
- Measure exact input variations and signal manifold dimension around arbitrary test points.

The primary contributions of this paper are theoretical in nature: establishing useful decompositions based on the exact representation theorem in Section 3 and writing several leading OOD detection methods in terms of this representation. The preliminary experimental results also support practical tasks of out-of-distribution (OOD) detection and estimating signal manifold dimension.

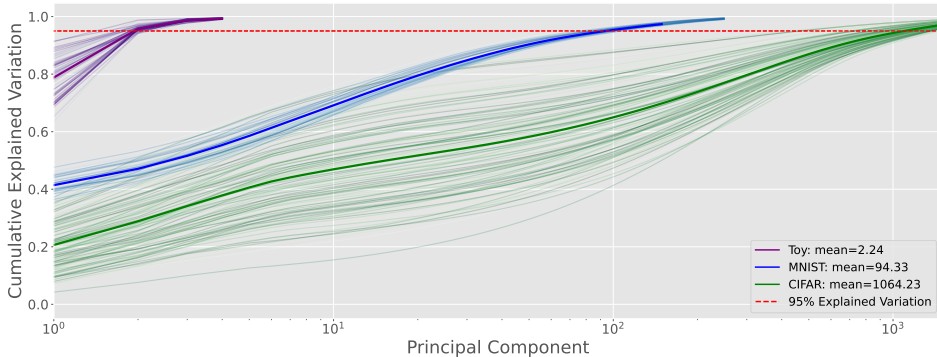

Figure 1: The gEPK naturally provides a measure of input dimension. This plot shows the CDF of the explained variation of training point sensitivities $\nabla_{x_\text{train}} f(x_\text{test}; \theta_\text{trained})$. Different datasets are color coded to show differences in signal dimension. Decomposing the input space in this way provides a view of the signal dimension around individual test points. For a toy problem (3 Gaussian distributions embedded in 100 dimensional space) the model only observes between 2 and 3 unique variations which contribute to 95% of the information required for prediction. Meanwhile the dimension of the signal manifold observed by the model around MNIST and CIFAR test points is approximately 94 and 1064 respectively.

## 2 RELATED WORK

While there has been a significant amount of recent work studying the Neural Tangent Kernel (NTK) (Jacot et al., 2018), there is still relatively little work exploring its exact counterpart, the path kernels (Bell et al., 2023; Chen et al., 2021b; Domingos, 2020). While these other works are focused on the precise equivalence between artificial neural networks and SVMs or Kernel machines, this equivalence requires significant restrictions placed on the loss function and model used for a task. This paper seeks to take advantage of this exact representation style without imposing such strict requirements. To the best of our knowledge, this is the first work exploring this loosened equivalence.

There are several schools of thought, whether OOD data can be learned (Huang & Li, 2021; Mohseni et al., 2020; He et al., 2015; Pillai et al., 2013; Fumera & Roli, 2002), which part of a model should be interrogated in order to identify OOD examples (Liu et al., 2020; Lin et al., 2021), whether it is a purely statistical question (Lee et al., 2018), or whether it can simply be solved with more data (Chen et al., 2021a; De Silva et al.). The best performing recent approaches have all used relatively simple modifications of model activation or model gradients (Djurisic et al., 2023; Xu et al., 2023; Sun & Li, 2022; Sun et al., 2021). The first methods we explore relates to the use of model gradients to construct statistics which separate in-distribution (ID) examples from OOD examples. This is fundamentally a geometric approach which should be comparable with the method proposed by Sun et al. (2022) (Gillette & Kur, 2022). The first prominent method of this type was proposed by Liang et al. (2018). ODIN is still a notable method in this space, and has been followed by many more gradient based approaches (Behpour et al., 2023; Huang et al., 2021b) and has caused some confusion about why these methods work so well (Igoe et al., 2022)

Much recent work has been devoted to measurement of dimension for the subspace in which the input data distribution live for machine-learning tasks. We will partition this work into works trying to understand this intrinsic data dimension in model agnostic ways (Gillette & Kur, 2022; Yousefzadeh, 2021; Kaufman & Azencot, 2023; Gilmer et al., 2018; Gong et al., 2019; Glielmo et al., 2022; Facco et al., 2018; Levina & Bickel, 2004) and works trying to understand or extract model's understanding of this subspace (Dominguez-Olmedo et al., 2023; Ansuini et al., 2019; Talwalkar et al., 2008; Costa & Hero, 2004b; Giryes et al., 2014; Zheng et al., 2022). This paper proposes a new method which bears more similarity to the latter. We believe that this approach is more relevant for studying ANNs since they discover their own metric spaces. Understanding signal manifolds is both useful in practice for more efficient low rank models (Yang et al., 2020; Swaminathan et al., 2020), and also for uncertainty quantification and robustness (Costa & Hero, 2004a; Wang et al., 2021; Khoury & Hadfield-Menell, 2018; Srinivas et al., 2023; Song et al., 2018; Snoek et al., 2019).

## 3   THEORETICAL JUSTIFICATION : EXACT PATH KERNEL DECOMPOSITION

The theoretical foundation of this starts with a modified general form of an recent exact path kernel representation result from Bell et al. (2023). We will reuse the structure of the Exact Path Kernel (EPK) without relying on the reduction to a single kernel across training steps. In order to increase generality, we will not assume the inner products may be reduced across steps, resulting in a representation which is no longer strictly a kernel. This representation however, will allow exact and careful decomposition of model predictions according to both input gradients and parameter gradients without the strict requirements of the EPK. The function, $\varphi_{s,t}(x)$, in the EPK sum defines a bilinear subspace, the properties of which we will study in detail. The primary difference between the representation we propose and the original EPK is the EPK maintained symmetry at the cost of continuity, on the other hand the gEPK does not introduce a discontinuity.

**Theorem 3.1** (Generalized Exact Path Kernel (gEPK)). *Suppose $f(\cdot; \theta) : \mathbb{R}^d \to \mathbb{R}^k$ is a differentiable parametric model with parameters $\theta_s \in \mathbb{R}^M$ and $L$ is a loss function. Furthermore, suppose that $f$ has been trained by a series $\{s\}_{s=0}^S$ of discrete steps composed from a sum of loss gradients for the training set $\sum_i^N \varepsilon \nabla_\theta L(f(x_i), y_i)$ on $N$ training data $X_T$ starting from $\theta_0$, with learning rate $\varepsilon$; as is the case with traditional gradient descent. Let $t \in [0, 1]$ be an interpolation variable which parameterizes the line connecting any $\theta_s$ to $\theta_{s+1}$ so that $\theta_s(t) = \theta_s + t(\theta_{s+1} - \theta_s)$. Then for an arbitrary test point $x$, the trained model prediction $f(x; \theta_S)$ can be written:*

$$f(x; \theta_S) = f(x; \theta_0) + \sum_{i=1}^N \sum_{s=1}^S \varepsilon \left( \int_0^1 \varphi_{s,t}(x) dt \right) L'(f(x_i; \theta_s), y_i) \left( \varphi_{s,0}(x_i) \right) \tag{1}$$

$$L'(a, b) = \frac{dL(a, b)}{db} \tag{2}$$

$$\varphi_{s,t}(x) \equiv \nabla_\theta f(x; \theta_s(t)), \tag{3}$$

$$\theta_s(t) \equiv \theta_s(0) + t(\theta_{s+1}(0) - \theta_s(0)), \text{ and} \tag{4}$$

$$\hat{y}_{\theta_s(0)} \equiv f(x; \theta_s(0)). \tag{5}$$

*Proof.* Guided by the proof for Theorem 6 from Bell et al. (2023), let $\theta$ and $f(\cdot; \theta)$ satisfy the conditions of Theorem 3.1, and $x$ be an arbitrary test point. We will measure the change in prediction during one training step from $\hat{y}_s = f(x; \theta_s)$ to $\hat{y}_{s+1} = f(x; \theta_{s+1})$ according to its differential along the interpolation from $\theta_s$ to $\theta_{s+1}$. Since we are training using gradient descent, we can write $\theta_{s+1} \equiv \theta_s + \dfrac{d\theta_s(t)}{dt}$. We derive a linear interpolate connecting these states using $t \in [0, 1]$:

$$\frac{d\theta_s(t)}{dt} = (\theta_{s+1} - \theta_s) \tag{6}$$

$$\int \frac{d\theta_s(t)}{dt} dt = \int (\theta_{s+1} - \theta_s) dt \tag{7}$$

$$\theta_s(t) = \theta_s + t(\theta_{s+1} - \theta_s) \tag{8}$$

One of the core insights of this definition is the distinction between *training steps* (defined by $s$) and the *path between training steps* (defined by $t$). By separating these two terms allows a *continuous* integration of the *discrete* behavior of practical neural networks. Since $f$ is being trained using a sum of gradients weighted by learning rate $\varepsilon$, we can write:

$$\frac{d\theta_s(t)}{dt} = -\varepsilon \nabla_\theta L(f(X_T; \theta_s(0)), y_i) \tag{9}$$

Applying chain rule and the above substitution, we can write the change in the prediction as

$$\frac{d\hat{y}}{dt} = \frac{df(x; \theta_s(t))}{dt} = \sum_{j=1}^M \frac{df}{\partial \theta^j} \frac{\partial \theta^j}{dt} = \sum_{j=1}^M \frac{df(x; \theta_s(t))}{\partial \theta^j} \left( -\varepsilon \frac{\partial L(f(X_T; \theta_s(0)), Y_T)}{\partial \theta^j} \right) \tag{10}$$

$$= \sum_{j=1}^M \frac{df(x; \theta_s(t))}{\partial \theta^j} \left( -\sum_{i=1}^N \varepsilon L'(f(x_i; \theta_s(0)), y_i) \frac{\partial f(x_i; \theta_s(0))}{\partial \theta^j} \right) \tag{11}$$

$$= -\varepsilon \sum_{i=1}^N \nabla_\theta f(x; \theta_s(t)) \cdot L'(f(x_i; \theta_s(0)), y_i) \nabla_\theta f(x_i; \theta_s(0)) \tag{12}$$

Using the fundamental theorem of calculus, we can compute the change in the model's output over step $s$ by integrating across $t$.

$$y_{s+1} - y_s = \int_0^1 -\varepsilon \sum_{i=1}^N \nabla_\theta f(x; \theta_s(t)) \cdot L'(f(x_i; \theta_s(0)), y_i) \nabla_\theta f(x_i; \theta_s(0)) dt \tag{13}$$

$$= -\sum_{i=1}^N \varepsilon \left( \int_0^1 \nabla_\theta f(x; \theta_s(t)) dt \right) \cdot L'(f(x_i; \theta_s(0)), y_i) \nabla_\theta f(x_i; \theta_s(0)) \tag{14}$$

For all $N$ training steps, we have

$$y_N = f(x; \theta_0) + \sum_{s=1}^N y_{s+1} - y_s \tag{15}$$

$$= f(x; \theta_0) - \sum_{s=1}^N \sum_{i=1}^N \varepsilon \left( \int_0^1 \nabla_\theta f(x; \theta_s(t)) dt \right) \cdot L'(f(x_i; \theta_s(0)), y_i) \nabla_\theta f(x_i; \theta_s(0)) \tag{16}$$

$\square$

**Remark 1:** While this theorem is not our main contribution, we provide it along with its brief proof to provide a thorough and useful theoretical foundation for the main results which follow.

**Remark 2:** Many of the remarks from Bell et al. (2023) remain including that this representation holds true for any contiguous subset of a gradient based model, e.g. when applied to only the middle layers of an ANN or only to the final layer. This is since each contiguous subset of an ANN can be treated as an ANN in its own right with the activations of the preceding layer as its inputs and its activations as its outputs. In this case, the training data consisting of previous layer activations may vary as the model evolves. One difference in this representation is that we do not introduce a discontinuity into the input space. This sacrifices symmetry, which disqualifies the resulting formula as a kernel, but retains many of the useful properties needed for OOD and dimension estimation.

**Remark 3:** Equation 16 allows decomposition of predictions into an initial (random) prediction $f(x; \theta_0)$ and a *learned adjustment* which separates the contribution of every training step $s$ and training datum $i$ to the prediction.

## 4 OOD IS ENABLED BY PARAMETER GRADIENTS

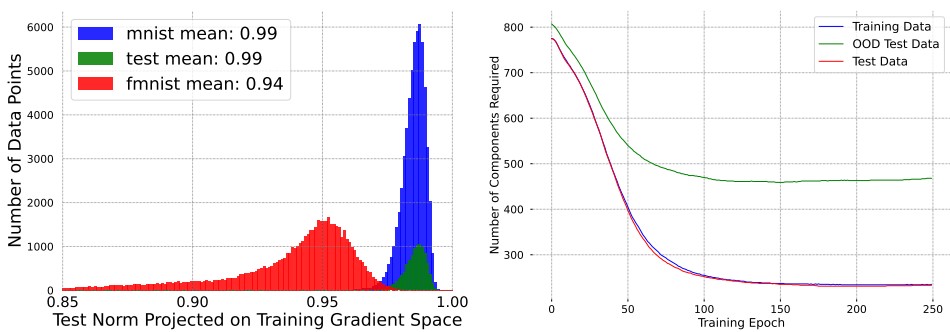

Figure 2: OOD detection using difference in training vs. test gradients. As the purpose of this paper is not to develop state of the art OOD detection methods, a comparison with recent benchmarks is not provided. Instead, a proof of concept that the gEPK can perform OOD detection is given. Left histogram shows norms of vectors projected onto the gradient weight space defined by the gEPK on MNIST and FMNIST. Right plot shows the number of components required to explain 95% variation in weight space across training for a toy problem (three Gaussian distributions embedded in 100 dimensions).

One natural application of the gEPK is the separation of predictions into vectors corresponding with the test gradient $\varphi_{s,t}(x)$ for a given test point $x$ and each training vector weighted by its loss gradient

$\frac{dL(\hat{y}_i, y_i)}{d\hat{y}_i}\varphi_{s,0}(x_i)$. While the test vector depends on the choice of test point $x$, the subspace of training gradient vectors is fixed. By the linear nature of this inner product, it is clear that no variation in test data which is orthogonal to the training vector space can be reflected in a model's prediction. We can state this as a theorem:

**Theorem 4.1** (Prediction Spanning Vectors).

$$B = \{\varphi_{s,0}(x_i); i \in \{1, ..., N\}, s \in \{1, ..., S\}\} \tag{17}$$

*spans the subspace of test parameter gradients with non-zero learned adjustments.*

*Proof.* Suppose for every $s$ and $t$, $\varphi_{s,t}(x) \notin B$. Then for every $i$, $s$, and $t$, $\langle \varphi_{s,t}(x), \varphi_{s,0}(x_i) \rangle = 0$. Rewriting equation 16 we have:

$$y_N = f(x; \theta_0) - \sum_{s=1}^{N} \sum_{i=1}^{N} \varepsilon \int_0^1 L'(f(x_i; \theta_s(0)), y_i) \langle \varphi_{s,t}(x), \varphi_{s,0}(x_i) \rangle dt \tag{18}$$

We can immediately see that every term in the learned adjustment summation will be equal to zero. $\square$

We will demonstrate that most cutting-edge OOD methods implicitly analyze the spectra of parts of this subspace in order to discriminate in practice.

## 4.1 Expressing Prior OOD Methods with the gEPK

We will now establish that most gradient based methods for OOD and some methods which do not explicitly rely on gradients can be written as projections onto subsets of this span.

**GradNorm** The first well-known method to apply gradient information for OOD is ODIN: Out-of-DIstribution detector for Neural Networks Liang et al. (2018). This method, inspired by adversarial attacks, perturbs inputs by applying perturbations calculated from input gradients. The method then relies on the difference in these perturbations for in-distribution versus out-of-distribution examples to separate these in practice. This method directly inspired Huang et al. (2021a) to create GradNorm. This method which occupied the cutting edge in 2021 computes the gradient of Kullback–Leibler divergence with respect to model parameters so that:

$$\frac{1}{C} \sum_i^C \frac{\partial L_{CE}(f(x; \theta), i)}{\partial \hat{y}} \nabla_\theta f(x; \theta) \tag{19}$$

This looks like the left side of the inner product from the gEPK, however the scaling factor, $\frac{\partial L_{CE}(f(x; \theta), i)}{d\hat{y}}$, does not match. In fact, this approach is averaging across the parameter gradients of this test point with respect to each of its class outputs, which we can see is only a related subset of the full basis used by the model for predictions. This explains improvements made in later methods that are using a more full basis. Another similar method, ExGrad (Igoe et al., 2022), has been proposed which experiments with different similar decompositions and raises some questions about what is special about gradients in OOD – we hope our result sheds some light on these questions. Another comparable method proposed by Sun et al. (2022) may also be equivalent through the connection we establish below in Section 1 between this decomposition and input gradients which may relate with mapping data manifolds in the Voronoi/Delaunay (Gillette & Kur, 2022) sense.

**ReAct, DICE, ASH, and VRA** Along with other recent work (Sun et al., 2021; Sun & Li, 2022; Xu et al., 2023), some of the cutting edge for OOD as of early 2023 involves activation truncation techniques like that neatly described by Djurisic et al. (2023). Given a model, $f(x; \theta) = f^{\text{extract}}(\cdot; \theta_{\text{extract}}) \circ f^{\text{represent}}(\cdot; \theta_{\text{represent}}) \circ f^{\text{classify}}(\cdot; \theta_{\text{classify}})$, and an input, $x$, a prediction, $f(x; \theta)$, is computed forward through the network. This yields a vector of activations, $A(x; \theta_{\text{represent}})$, in the representation layer of the network. This representation is then pruned down to the $p^{\text{th}}$ percentile by setting any activations below that percentile to zero. Djurisic et al. (2023) mention that ASH does not depend on statistics from the training data, however by chain rule, high activations will correspond with high parameter gradients. Meaning this truncation is picking a representation

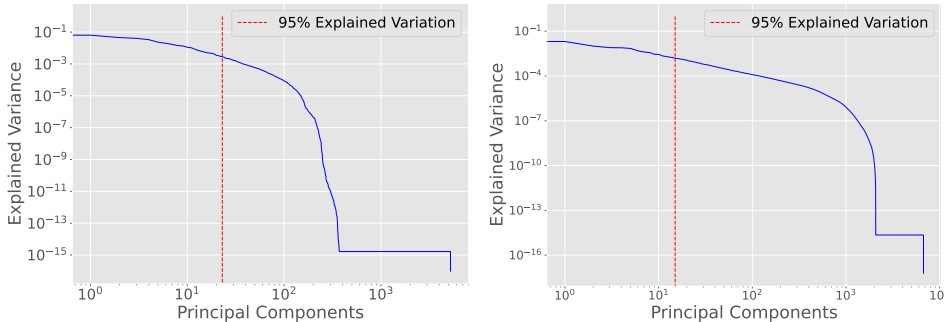

Figure 3: Explained Variance Ratio of parameter gradients. Left: MNIST, Right: CIFAR. 95% of variation can be explained with a relatively low number of components in both cases.

for which $\left\langle \nabla_\theta f(x; \theta_{\text{represent}}), \dfrac{dL(\hat{y}(x_i), y_i)}{d\hat{y}} \nabla_\theta f(x_i; \theta_{\text{represent}}) \right\rangle$ is high for many training points, $x_i$.

This is effectively a projection onto the parameter tangent space of the training data with the highest variation. This may explain some part of the performance advantage of these methods.

**GradOrth** Behpour et al. (2023) explicitly create a reference basis from parameter gradients on training data for comparison. They do this for only the last layer of a network with mean squared error (MSE) loss, allowing a nicely abbreviated expression for the gradient:

$$\nabla_\theta L(x, y) = (\theta x - y)x^T = \Omega x^T \tag{20}$$

Treating $\Omega$ as an error vector, they prove that all variation of the output must be within the span of the $x^T$ over the training set. They then pick a small subset of the training data and record its activations $R_{ID}^L = [x_1, x_2, ..., x_n]$ over which they compute the SVD, $U_{ID}^L \Sigma_{ID}^L (V_{ID}^L)^T = R_{ID}^L$. This representation is then truncated to $k$ principal components according to a threshold $\epsilon_{\text{th}}$ such that

$$\left\| U_{ID}^L \Sigma_{ID,k}^L (V_{ID}^L)^T \right\|_F^2 \geq \epsilon_{\text{th}} \| R_I^L D \|_F^2. \tag{21}$$

This basis $S^L = (U_I^L D)_k$ is now treated as the reference space onto which test points' final layer gradients can be projected. Their score is:

$$O(x) = (\nabla_{\theta_L} \mathcal{L}(f(x; \theta_L), y)) S^L (S^L)^T \tag{22}$$

We note that this formulation requires a label $y$ for each of the data being tested for inclusion in the data distribution. Despite this drawback, the performance presented by Behpour et al. (2023) is impressive.

## 4.2 GEPK FOR OOD

Theorem 4.1 provides a more general spanning result immediately. In fact, as we have illustrated in Figure 2, we can pick a much reduced basis *based only on the final training step* which will span most of the variation in models' learned adjustments. Theorem 4.1 and the definition of SVD provide the following:

**Corollary 4.2.** *Let $A$ be a matrix stacking the elements of $B$ as rows. Then let $U\Sigma V^T = A$ as in SVD. Then $Span(B) = Span(Rows(V))$.*

In the case that the number of training data exceed the number of parameters of a model, the same result holds true for a basis computed only for gradients with respect to the final parameter states $\theta_S$. We will use a truncation, $V'$ of this final training gradient basis which we examine in Fig. 3. This truncation still explains most variation in all layers due to the convergence of training gradients to a smaller subspace as shown in Fig. 2. In future it may be possible to argue statistical expectations about the performance of a sketching approach to producing an equally performant basis without expensive SVD.

We can see that most, if not all, of the above OOD methods can be represented by some set of averaging or truncation assumptions on the basis $V$. These should be mostly caught by the truncated

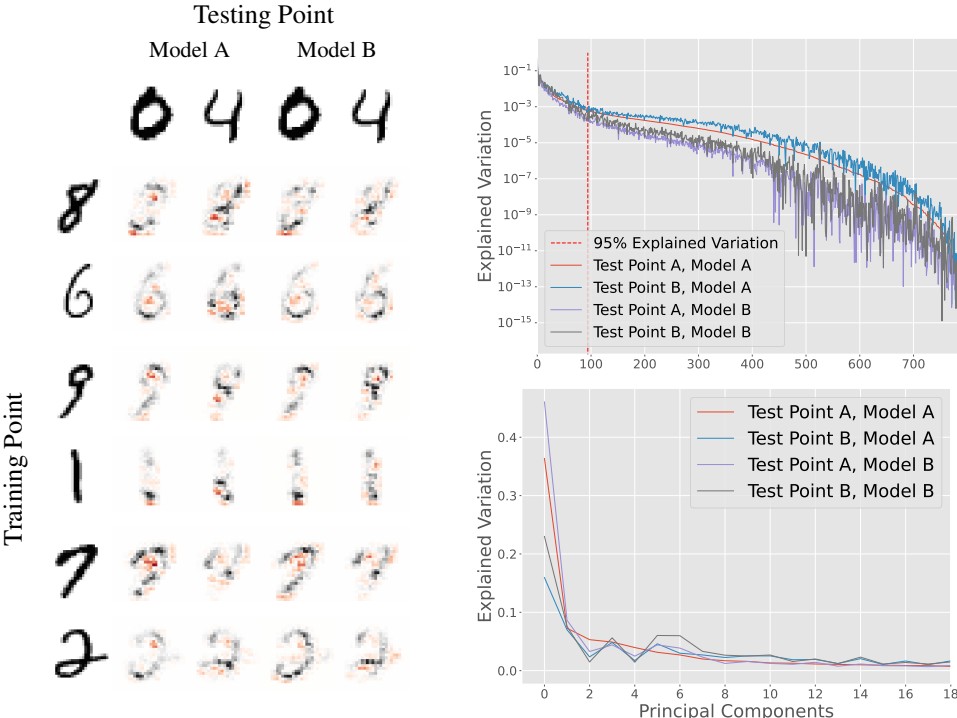

Figure 4: Left: Visualization of training point input gradients on test points compared between two models. Positive contribution (black) and negative contribution (red) of each training datum to the prediction for each test point. Elements in the grid are $\nabla_{x_{\text{train}}} f(x_{\text{test}}; \theta_{\text{trained}})$. Right: By taking these individual gradient contributions for a test point and computing the SVD across the training, the significant modes of variation in the input space can be measured (sigma squared). Top is log scale of the full spectrum, bottom shows the first 10 components. Note that this decomposition selects similar, but not identical, modes of variation across test points and even across different models. Components in SVD plots are sorted using Test Point A on Model A.

basis $V$. We test the usefullness of $V'$ to perform OOD detection by projection onto its span using a sum over the class outputs weighted by the loss gradients $L'(f(x_i; \theta_S), y_i)$ in Fig. 2. We note that this scalling has only been extracted from the final training step, however this assumption is supported by the convergence of this scaling over training. Indeed, this helps explain the high performance of gradient based methods due to the implicit inclusion of the training parameter space in model predictions. This serves to illuminate the otherwise confusing discrepancy raised by Igoe et al. (2022).

In addition, we can see that comparison of test versus training loss gradients is unnecessary, which allows testing on data without ground truth labels (an issue with many recent gradient based OOD techniques). For most applications, the SVD of the parameter gradients over all of the training steps and batches can be pre-computed and compared with test points as needed, although as we can see from this body of work, many simplifying assumptions can be made which will preserve the essential bases needed for performance, but still drastically reduce computational cost. Bottom line: It is not necessarily sufficient to pick a basis that spans a target subspace and then truncate based on its variations. The variations must be accurately measured with correct scaling in terms of their contribution to the learned adjustments of a model.

## 5 SIGNAL MANIFOLD DIMENSION ESTIMATED WITH TRAINING INPUT GRADIENTS

In order to understand the subspace on which a model is sensitive to variation, we may take gradients decomposed into each of the training data. Take, for example, a model, $f(x; \theta)$, which satisfies the

necessary conditions for expression as:

$$f(x; \theta_{\text{trained}}) = f(x; \theta_0(0)) + \sum_i \sum_s \int_0^1 \varphi_{s,t}(x) \frac{dL(x_i, y_i)}{df(x_i; \theta_s(0))} \varphi_{s,0}(x_i) dt \tag{23}$$

$$\varphi_{s,t}(x) = \nabla_\theta f(x; \theta_s(t)) \tag{24}$$

And $\theta_s(t)$ are the parameters of $f$ for training step $s$ and time $t$ so that $\sum_s \int_0^1 \theta_s(t) dt$ integrates the entire training path taken by the model during training. Given a test point $x$, we can evaluate its subspace by taking, for each $x_i$:

$$\frac{df(x; \theta_{\text{trained}})}{dx_j} = \frac{df(x; \theta_0(0))}{dx_j} + \sum_i \sum_s \int_0^1 \frac{d\left(\varphi_{s,t}(x) \frac{dL(x_i, y_i)}{df(x_i; \theta_s(0))} \varphi_{s,0}(x_i)\right)}{dx_j} dt \tag{25}$$

$$= \sum_i \sum_s \int_0^1 \varphi_{s,t}(x) dt \left(\frac{d^2 L(x_i, y_i)}{df(x_i; \theta_s(0)) dx_j} \varphi_{s,0}(x_i) + \frac{dL(x_i, y_i)}{df(x_i; \theta_s(0))} \frac{d\varphi_{s,0}(x_i)}{dx_j}\right) \tag{26}$$

We can see that these gradients will be zero except when $i = j$, thus we may summarize these gradients as a matrix (tensor in the multi-class case), $G$, with

$$G_j = \sum_s \int_0^1 \varphi_{s,t}(x) dt \left(\frac{d^2 L(x_i, y_i)}{df(x_i; \theta_s(0)) dx_j} \varphi_{s,0}(x_i) + \frac{dL(x_i, y_i)}{df(x_i; \theta_s(0))} \frac{d\varphi_{s,0}(x_i)}{dx_j}\right) \tag{27}$$

While written in this form, it appears we must keep second-order derivatives, however we note that the inner product with $\phi_{s,t}(x)$ eliminates these extra dimensions, so that clever implementation still only requires storage of vectors (low rank matrices in the multi-class case).

The rank of $G$ represents the dimension of the subspace on which the model perceives a test point, $x$, to live, and we can get more detailed information about the variation explained by the span of this matrix by taking its SVD. We can exactly measure the variation explained by each orthogonal component of the $\text{span}(G)$ with respect to the given test point $x$. $G(x)$ can be defined as a map from $x$ to the subspace perceived by the model around $x$. Any local variations in the input space which do not lie on the subspace spanned by $G(x)$ can not be perceived by the model, and will have no effect on the models output.

On MNIST, $G(x)$ creates a matrix which is of size $60000 \times 784 \times 10$ (training points $\times$ input dimension $\times$ class count). This matrix represents the exact measure of each training points contribution towards a given test prediction. In order to simplify computation, we reduce this term to $60000 \times 784$ by summing across the class dimension. This reduction is justified by the same theory as the psudo-NTK presented by Mohamadi et al. (2023). Of note is that in practice this matrix is full rank on the input space as seen in Figure 4. This is despite MNIST having significantly less degrees of variation than its total input size (many pixels in input space are always 0). Figure 1 demonstrates that accounting for 95% of the variation requires only 94 (12%) of the 784 components on average. Similarly, on CIFAR accounting for 95% of explained variation requires 1064 (34%) of the 3096 components. It is likely that different training techniques will provide significantly different signal manifolds and consequently different numbers of components. We can also examine this subspace with less granularity by taking the parameter gradients for each training point from its trained state. This involves using each training point as a test point.

$$\frac{df(x_j; \theta_{\text{trained}})}{dx_j} = \frac{df(x_j; \theta_0(0))}{dx_j} + \sum_i \sum_s \int_0^1 \frac{d\left(\varphi_{s,t}(x_j) \frac{dL(x_i, y_i)}{df(x_i; \theta_s(0))} \varphi_{s,0}(x_i)\right)}{dx_j} dt \tag{28}$$

The left hand side is computable without path-decomposition and so can be computed for each training datum to create a gradient matrix, $H_{\theta_{\text{trained}}}$. Another term, $\frac{df(x_j; \theta_0(0))}{dx_j}$ is also easily computable, yielding another matrix $H_{\theta_0}$. By comparing the rank and span of $H_{\theta_{\text{trained}}}$ and $H_{\theta_0}$ we can understand to what extent the model's spatial representation of the data is due to the initial parameter selection and how much is due to the training path. Also, $H_{\theta_{\text{trained}}}$ provides sample of gradients across all

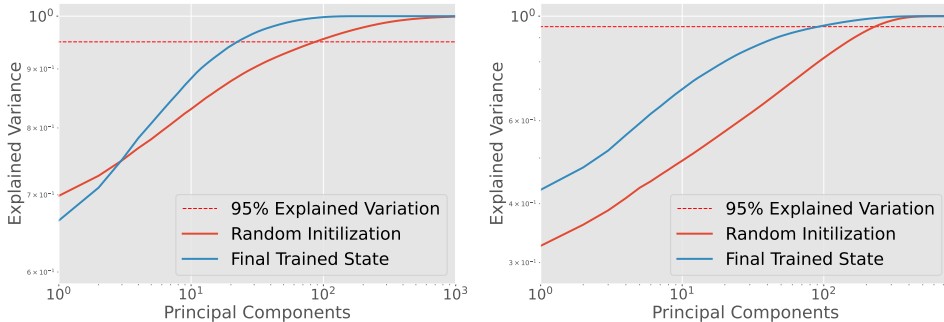

Figure 5: Differences between observing gradients in input space vs. weight space. Left: CDF of explained variation parameter space. Right: CDF of explained variation input space. Red solid line indicates a model at random initialization while the blue solid line represents the fully trained state. From random initialization, the number of principal components required to achieve 95% explained variation decreases in both cases. Note that at random initialization, the weight space gradients already have only a few directions accounting for significant variation. Disentangling the data dimension using weight space gradients is less effective than doing so in input space (Shamir et al., 2021).

training data, which in some sense must be spanned by the model's implicit subspace basis. Despite missing the granular subspace information, the rank of this gradient matrix and its explained variation computed using SVD should be related to the model's implicit subspace rank. It should be noted that while there is a direct relationship between a models variations in input space and weight space, Figure 5 shows that this mapping changes greatly from the beginning to end of training and that this spectrum starts out wide (high dimensional) for $\theta_0$ and much more focused (low dimensional) for $\theta_T$.

One interesting property of using input gradients for training data decomposed according to equation 27 is the ability to compare input gradients across models with different initial parameters and even different architectures. Figure 4 demonstrates that two models with different random initializations which have been trained on the same dataset have a signal manifold which shares many components. This is a known result that has been explored in deep learning through properties of adversarial transferability Szegedy et al. (2013). This demonstrates that the gEPK is capable of measuring the degree to which two models rely on the same features directly. This discovery may lead to the construction of models which are provably robust against transfer attacks.

## 6 CONCLUSION

This paper presented decompositions based on a general exact path kernel representation for neural networks with a natural decomposition that connects existing out-of-distribution detection methods to a theoretical baseline. This same representation reveals additional connections to dimension estimation and adversarial transferability. These connections are demonstrated with experimental results on computer vision datasets. The key insights provided by this decomposition are that model predictions implicitly depend on the parameter tangent space on its training data and that this dependence enables decomposition relative to a single test point by either parameter gradients, or training input gradients. This allows users to connect how neural networks learn at training time with how each training point influences the final decisions of a network. We have demonstrated that the techniques used in practice for OOD are using a subset of the theoretical basis we propose. Taking into account the entire training path will allow more rigorous methods for OOD detection. There are many possible directions to continuing work in this area. These include better understanding of how models depend on implicit prior distributions following (e.g. Nagler (2023)), supporting more robust statistical learning under distribution shifts (e.g. Simchowitz et al. (2023)), and supporting more robust learning.

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
