# OpenReview forum: "Exact Path Kernels Naturally Decompose Model Predictions"
_ICLR.cc/2024/Conference — Submitted to ICLR 2024_

### Official Review · Reviewer_dZQY · 2023-11-01

**Soundness:** 2 fair
**Presentation:** 2 fair
**Contribution:** 2 fair
**Rating:** 5
**Confidence:** 2

**Summary:**

The paper presents a generalized exact path kernel as well as its application in out-of-distribution detection.

**Strengths:**

1. The idea of the approach is intuitive and simple: the authors first establish the connection between the model prediction and the gradient of the loss function on the training data, then they perform spectral analysis on the subspace formed by the gradient vectors.
2. The proposed generalized exact path kernel provides a natural interpretation of the prediction in terms of the loss and parameter gradients.
3. Empirical results demonstrate the effectiveness of the method in identifying the signal manifold dimension.

**Weaknesses:**

1. Presentation of the manuscript can be improved: Some notations are not clearly defined. For example, \epsilon in Theorem 3.1 is not introduced but only mentioned in the proof as a constant. Similarly, \theta_s(t) is not defined properly and is not distinguished from \theta_s. From the context, s denotes the iteration and t denotes the training data index, does \theta_s(t) refer to a parameter specific to data point t?
2. Theorem 3.1 is the foundation of the proposal but the proof does not seem to be rigorous: it states that θ_{s+1} ≡ θ_s + dθ_{s(t)} / dt, is a step size missing there?
3. The OOD is obtained by analyzing the span of the linear subspace formed by the loss gradient vectors. However, the analysis may not take into account the nonlinearity.

**Questions:**

- What's the significance of \epsilon in Theorem 3.1 and how is it determined?

---

> ### Author Response · Authors · 2023-11-16
> **Response to reviewer comments**
>
> Thank you for reviewing our manuscript and providing your insightful critiques. We appreciate you taking the time to thoroughly review our work.
>
> In response to your first concern, you are correct, a definition for $\varepsilon$ was missing. In this case $\varepsilon$ represents the learning rate hyperparameter of the neural network and we have added a comment defining it as such to the definitions in Theorem 3.1.
> In regards to $\theta_s(t)$ the definition of this value can be found in section 3 equation 7 and the relevant properties of $\theta_s(t)$ as it relates to $\theta_s$ are defined in equations 5 and 6. We agree that the notation of this section may be confusing, so we have clarified the domain of $t$ and added a section explaining the properties of $\theta_s(t)$.
>
> We understand your concerns about the limitations of linear methods to address the nonlinear nature of neural networks. We have substantially revised section 4 in order to discuss these details in a clearer way.  In both of our applications, the nonlinearity of the model is contained in how the model spatially transforms and represents the data for the task. Both in parameter space and in input space, this generates nonlinear (curved) spatial arrangements of data. Although this decomposition can be used to analyze nonlinearity, our applications to both OOD and dimension measurement are limited to the tangent space around specific individual data. In this context, the tangent space is linear. Our revised section 4 addresses the implications of our choice to pick a linear basis for the final trained state and extend that basis across all training steps. An exact decomposition of a prediction into contributions for each training point requires a non-linear basis across the space of training paths. However, since the training path is fixed, we can still explain all variation by only looking at the linear subspaces for each training step. This is subtle but useful and relies on the fact that the training gradients in this representation do not depend on $t$. We justify reducing this down to only a truncated basis for the final training step after Corollary 4.2. We hope that these revisions address your concerns and questions, and we greatly value your continued feedback to continue clarifying these subtle details.
>
> We sincerely appreciate you taking the time to thoughtfully review our work. Your insights have undoubtedly helped us strengthen this manuscript.

---

### Official Review · Reviewer_HDfE · 2023-11-01

**Soundness:** 3 good
**Presentation:** 1 poor
**Contribution:** 2 fair
**Rating:** 5
**Confidence:** 3

**Summary:**

The paper offers valuable insights into the concept of exact path kernels, enabling us to interprete prediction through the utilization of input gradients over training trajectories. This methodology was first introduced by Bell et al. 2023. In addition, the paper extends the approach to provide a more generalized representation of the output of a test point under general loss functions. Moreover, it establishes a connection between exact path kernels and out-of-distribution (OOD) methods such as GradNorm, GradOrth, ReAct, DICE, and others, all of which rely on the utilization of parameter gradients. The paper also delves into the exploration of using a modified Singular Value Decomposition (SVD) to estimate the signal manifold.

**Strengths:**

- The paper gives several interesting insights of exact path kernels in the setting of OOD detection.
- Signal manifold estimation seems interesting.

**Weaknesses:**

- The paper writing can be improved, i.g. punctuation for equations.
- I find that there is repetition like Bell et al. on Section 3, remark 1 and remark 2.That is, Bell et al. make the similar remarks on SGD and subsampling. Proving techniques resembles the approach in Bell et al.,
- While EPK gives a representation of test points in terms of a vector space where the basis is computed from gradients at every training step, most OOD methods consider the optimized parameter which is $\theta_S$ as the notation in the paper. This means that the existing OOD does not consider the parameter trajectories of optimization.

**Questions:**

Minor points
- $L_{CE}$ is not defined.

---

> ### Author Response · Authors · 2023-11-16
> **Response to reviewer comments**
>
> Thank you for your thoughtful feedback on our manuscript. We appreciate you taking the time to provide an in-depth review and value your expertise in improving the quality of our work.
>
> We understand your concerns about the limitations of the gEPK and its similarity to prior work. Per your recommendations, we have made revisions to improve this aspect in the beginning of section 3. We now more clearly indicate the differences between work done previously by Bell et al. and our paper. In addition we have revised Section 4 in order to provide more detail on the novelty of our approach.
>
> Regarding your concern of differences between accounting for the training path or evaluating only at the final trained state, we have addressed this issue in the revised section 4. You will now find Theorem 4.1 which refers to the most general basis for all training parameter gradients along the path and Corollary 4.2 which discusses the use of a reduced basis and discusses a justification for using only the final training state. This provides a more rigorous foundation for gradient based OOD by taking into account this full decomposition, and also places many cutting edge OOD approaches in a more rigorous and interpretable context by highlighting their relation (as truncations or as implicit subspaces) to the subspaces and basis we define. There is more work to be done on this in order to bring it into a practical setting. In response to your concern we have added more discussion surrounding this topic to the conclusion of this paper.
>
> Thank you again for your help in improving the quality of this work. If you have any further questions or concerns we would be happy to continue this discussion.

---

### Official Review · Reviewer_eT4X · 2023-11-01

**Soundness:** 2 fair
**Presentation:** 2 fair
**Contribution:** 2 fair
**Rating:** 3
**Confidence:** 2

**Summary:**

This paper provides a decomposition of the prediction of differentiable models in terms of parameter gradients throughout training. It is shown that a subspace arising from this decomposition captures a significant portion of model prediction variations at a test point when perturbing training points. Some links are established with out-of-distribution detection methods and measuring the dimension of the signal manifold.

**Strengths:**

There are interesting connections between the provided decomposition and OOD/signal manifold dimension detection methods, which might be worth further exploration.

**Weaknesses:**

* My main concern is that the theoretical results seem to follow directly from the arguments of Bell et al., 2023. Perhaps it could help if the authors added a discussion on the technically novel aspects of the analysis, or alternatively emphasized more on the applications of this decomposition as the main contribution.

* While I am not an expert in OOD detection and thus cannot assess the work based on its applied contributions, I am wondering if it is possible to have experiments where additional intuitions gained from the analysis of this work concretely improve the performance of some methods.

* There is some notational ambiguity and mathematical imprecision in the current version of the manuscript. Specific examples are provided in the questions section below.

**Questions:**

* Some examples for improving the clarity and rigor of the mathematical statements:
    * It might be more appropriate to write Eq. (1) in terms of the inner product notation $\langle \varphi_{s,t}(x), \varphi_{s,0}(x_i)\rangle$.
    * It could help the readers better understand the setting if the authors clarify the meaning of "$f$ has been trained by a series of discrete steps composed from a sum of loss gradients ...". If this is just training via gradient descent, it might be easier to simply write down the GD equation.
    * The input and output space of $f$ don't seem to be defined.
    * Making sense of Eq. (8) is not straightforward as the LHS is a vector in $\mathbb{R}^m$, while the RHS is a summation over $M$ scalars (if $\theta^j$ is supposed to denote the $j$th index of $\theta$).
    * The notation $\frac{dL(f(x_i,\theta_s),y_i)}{d\hat{y}_{\theta_s(0)}}$ does not seem to be immediately interpretable. If my understanding of the meaning of this expression is correct, my suggestion is to define something like $L'(\cdot,\cdot)$ as the derivative of $L(\cdot,\cdot)$ with respect to its first argument, and use $L'(f(x_i,\theta_s(0)),y_i)$ instead.

---

> ### Author Response · Authors · 2023-11-16
> **Response to reviewer comments**
>
> Thank you for taking the time to review our manuscript and provide your feedback. We greatly appreciate you sharing your expertise to help strengthen our work. We have carefully considered all of the suggestions you provided in your review. In response, we have made the following changes to the manuscript:
>
>  - Substantially revised Section 4 for clarity and to highlight our contributions to the detailed theory needed to use this method to decompose projections.
>  - Added a description to Theorem 3.1 indicating that these constraints are satisfied by gradient descent
> -  Added definitions for the input and output space of the network function $f$
>  - Removed the RHS of equation (8) as it was indeed in error and could cause confusion
>  - Simplified the notation surrounding the loss gradient to improve the clarity of our equations
>  - Clarified the definition of $\epsilon$ in Theorem 3.1 to be the learning rate of the neural network
>
> The focus of this paper is on the use of a generalized exact path kernel to decompose predictions. In order to present this theoretical framework in a readable, complete, and transparent way, we have included a proof of the generalized representation which is derived from Bell et al. This is not our main contribution. Our main theoretical contribution is in the application of this representation to decompose predictions and the consideration of many details including how this decomposition applies across training steps and model layers. We have presented these details alongside applications to OOD and manifold dimension measurement to demonstrate the applicability and usefulness of this approach. While we agree that building this analysis into a framework that performs OOD detection and testing it on benchmark datasets is of value; this is a significant task which could stand on its own as a contribution to a more application focused venue, and so we consider it future work.
>
> We would be happy to provide any additional information needed during the next steps of the review process. Thank you again for your time and for helping enhance our research. We greatly appreciate the effort you have put into reviewing our manuscript.

---

> > ### Comment · Reviewer_eT4X · 2023-11-22
> >
> > Thank you for your detailed response and for revising the manuscript. I believe that the provided decomposition can have interesting applications including OOD detection and manifold dimension estimation as presented by the authors. However, I believe the current work can be significantly strengthened if it can either show more formal connections and comparisons with theoretical guarantees between recent OOD methods through this decomposition, or can use the intuitions to come up with modifications to current methods that provide meaningful improvements. These issues are perhaps best addressed in a new submission. For this reason, I am keeping my original score.

---

### Meta-Review · Area_Chair_SsSj · 2023-12-03

**Metareview:**

The paper generalizes the Exact Path Kernel, first proposed in Bell et al. (2023). The resulting kernel, called gEPK, allows for the decomposition of predictions in terms of the gradients for a given test point and training data. The results have implications for out-of-distribution detection methods and measuring the dimensionality of the signal manifold.

Although the overall ideas in the paper are interesting, there are major concerns regarding the degree of novelty of the results. The contributions of the paper appear incremental in light of the prior works. The writing and presentation of the paper can also be significantly improved.

**Justification For Why Not Higher Score:**

The contributions of the paper appear incremental in light of the prior work. The writing and presentation of the paper can also be significantly improved.

**Justification For Why Not Lower Score:**

N/A

---

### Decision · Program_Chairs · 2024-01-16

Reject